# CCN1/Integrin α_5_β_1_ Instigates Free Fatty Acid-Induced Hepatocyte Lipid Accumulation and Pyroptosis through NLRP3 Inflammasome Activation

**DOI:** 10.3390/nu14183871

**Published:** 2022-09-19

**Authors:** Qinyu Yao, Jia Liu, Qi Cui, Tingting Jiang, Xinya Xie, Xiong Du, Ziwei Zhao, Baochang Lai, Lei Xiao, Nanping Wang

**Affiliations:** 1Cardiovascular Research Center, School of Basic Medical Sciences, Xi’an Jiaotong University, Xi’an 710061, China; 2Key Laboratory of Environment and Genes Related to Diseases, Xi’an Jiaotong University, Ministry of Education of China, Xi’an 710061, China; 3Advanced Institute for Medical Sciences, Dalian Medical University, Dalian 116044, China; 4Health Science Center, East China Normal University, Shanghai 200241, China

**Keywords:** CCN1, integrin α_5_β_1_, NLRP3 inflammasome, pyroptosis, hepatocytes

## Abstract

Hyperlipidemia with high blood levels of free fatty acids (FFA) is the leading cause of non-alcoholic steatohepatitis. CCN1 is a secreted matricellular protein that drives various cellular functions, including proliferation, migration, and differentiation. However, its role in mediating FFA-induced pro-inflammatory cell death and its underlying molecular mechanisms have not been characterized. In this study, we demonstrated that CCN1 was upregulated in the livers of obese mice. The increase in FFA-induced CCN1 was evaluated in vitro by treating hepatocytes with a combination of oleic acid and palmitic acid (2:1). Gene silencing using specific small interfering RNAs (siRNA) revealed that CCN1 participated in FFA-induced intracellular lipid accumulation, caspase-1 activation, and hepatocyte pyroptosis. Next, we identified integrin α_5_β_1_ as a potential receptor of CCN1. Co-immunoprecipitation demonstrated that the binding between CCN1 and integrin α_5_β_1_ increased in hepatocytes upon FFA stimulation in the livers of obese mice. Similarly, the protein levels of integrin α_5_ and β_1_ were increased in vitro and in vivo. Experiments with specific siRNAs confirmed that integrin α_5_β_1_ played a part in FFA-induced intracellular lipid accumulation, NLRP3 inflammasome activation, and pyroptosis in hepatocytes. In conclusion, these results provide novel evidence that the CCN1/integrin α_5_β_1_ is a novel mediator that drives hepatic lipotoxicity via NLRP3-dependent pyroptosis.

## 1. Introduction

Hyperlipidemia, manifesting with high blood levels of free fatty acids (FFA), is a leading cause of multiple metabolic diseases. Circulating levels of FFA increase markedly in individuals with type 2 diabetes, obesity, and non-alcoholic steatohepatitis (NASH) [1]. NASH is a progressive non-alcoholic fatty liver disease (NAFLD) involving excessive hepatic lipid accumulation, inflammation, and fibrosis [2]. The lipotoxicity of FFA is attributed to the release of pro-inflammatory cytokines and increased lipid accumulation and peroxidation. In addition to triggering metabolic inflammation, these events are thought to cause DNA damage and cell death.

Inflammasomes are multimeric complexes that assemble in response to infection or chronic sterile inflammation. Activation of the NOD-like receptor protein 3 (NLRP3), which is characterized by caspase-1 cleavage, can aggravate inflammation via both interleukin-1b (IL-1β) and IL-18. Aberrant or excessive activation of caspase-1 has a key role in lipid regulation and hepatocyte death upon pathogen attack [3]. Pyroptosis is a form of lytic programmed cell death, which detects intracellular infection or metabolic perturbation. It can be induced by NLRP3-triggered caspase-1 activation or by caspase-4/5/11, which is independent of the inflammasome. Gasdermin D (GSDMD) is one of the gasdermin family members that execute pyroptosis via their pore-forming activity. The caspases cleave GSDMD and separate its N-terminal pore-forming domain from the C-terminal repressor domain. The N-terminal fragment of the GSDMD (GSDMD-NT) product is inserted into the cell membrane to assemble arc- and slit-like oligomers and form transmembrane pores, leading to cell membrane rupture and resultant pyroptosis [4]. Recent studies have proved that activation of the NLRP3 inflammasome in hepatocytes promotes GSDMD-dependent pyroptotic cell death and consequent liver fibrosis [5]. Recently, hepatocyte pyroptosis has been studied in fatty liver disease models in vitro and in vivo. However, the molecular mechanisms underlying the effects of the pro-pyroptotic action remain largely unclear.

CCN1 (also known as CYR61), the first member of the CCN family, was identified as a cysteine-rich and a serum-inducible protein. Although CCN1 was initially thought to trigger cellular proliferation and embryonic development, critical roles in fibrosis [6], ischemia/reperfusion [7], and alcohol-induced liver injury [8] have been discovered in adults under pathological conditions. The induction of CCN1 protein by lipopolysaccharide (LPS) has been shown to occur in murine hepatocytes through the TLR4/MyD88/AP-1 pathway [9]. The CCN1 protein level, which is induced by IL-17 in a p38 MAPK and NF-κB-dependent manner, is elevated in synovial fluid samples from rheumatoid arthritis patients, as compared with healthy controls [10]. These studies have highlighted the induction of CCN1 as a critical response to infection or chronic inflammation. In contrast to classical growth factors secreted into the cell-cultured medium, CCN1 was found to be tightly associated with the extracellular matrix and the cell surface. The most documented CCN-binding receptors are integrins, including α_V_β_3_, α_V_β_5_, α_M_β_2_, and α_D_β_2,_ which mediate diverse functions in various cell types [11]. However, the integrin heterodimer responsible for CCN1-induced inflammatory pyroptosis remains to be characterized. Hepatic CCN1 has been shown to be positively correlated with steatosis in patients with NASH [12]. It also induces macrophage infiltration in the livers of high fat diet (HFD)-fed mice [9], and increases intracellular triglyceride contents and apoptosis-associated proteins in primary hepatocytes [12]. However, the involvement of CCN1 in pyroptosis has not been described.

Integrins are heterodimeric transmembrane proteins consisting of 24 non-covalently associated a and b subunits. They provide the central mechanism by which cells sense and interact with their extracellular environment [13]. CCN1 binds and signals through cell surface integrins, such as integrin α_6_β_1_, α_V_β_3_, and α_V_β_5_, to regulate a wide range of cellular processes, including proliferation, survival, and migration [14]. Several studies have implicated integrins in physiopathological processes of the liver, including monocyte/macrophage adhesion [15], fibrosis [16], and the development of hepatocellular carcinoma [17]. Blocking integrin α_4_β_7_ suppresses T cell recruitment to the liver, thereby protecting mice from Western diet-induced NASH [18]. LPS-generated integrin β_1_ vesicles favor monocyte adhesion to liver sinusoidal endothelial cells in a murine NASH model [15]. Recent research indicates that integrin α_D_-deficient neutrophils exhibit increased necrosis and pyroptosis, resulting in reduced lung inflammation and mortality [19]. However, the implications for hepatic inflammatory cell death have not been described. In addition, the integrin heterodimer responsible for CCN1 signaling transduction remains to be characterized.

In this study, we identified integrin α_5_β_1_ as a potential receptor of CCN1 in hepatocytes. We also examined the role of the CCN1/integrin α_5_β_1_ in hepatocyte inflammation, intracellular lipid accumulation, and pyroptosis.

## 2. Materials and Methods

### 2.1. Cell Culture

Human hepatocyte LO2 cells were obtained from ATCC (Manassas, VA, USA) and cultured in Dulbecco’s modified Eagle’s medium supplemented with 10% fetal bovine serum.

### 2.2. Animal Procedures

Male C57BL/6 mice were bred under controlled temperature conditions (25 °C), with a 12 h light and dark cycle. Animal experiments were approved by the Institutional Ethics Review Board of Xi’an Jiaotong University (XJTULAC2015-404) and performed in accordance with the NIH guidelines for the care and use of animals. Mice at 8 weeks old were fed a HFD (60% fat) or kept on a normal diet (ND), (Research Diets, New Brunswick, NJ, USA) for 12 weeks. Male obese db/db and db/m mice were raised to 8 weeks old.

### 2.3. Reagents

Primary antibodies against integrin α_5_, NLRP3, caspase-1, GSDMD, high mobility group box 1 (HMGB1), and CCN1 were acquired from Cell Signaling (Danvers, MA). Antibodies against integrin β_1_ and β-actin were obtained from Santa Cruz Biotechnology (Dallas, TX). Bovine serum albumin (BSA), oleic acid (OA), and palmitic acid (PA) were received from Sigma-Aldrich (St. Louis, MO, USA). To prepare the FFA mixture, OA and PA (2:1) were emulsified in phosphate-buffered saline (PBS) containing 1% BSA.

### 2.4. Transfection

LO2 hepatocytes were transfected with small interfering RNAs (siRNAs) against CCN1, integrin α_5_ or integrin β_1_, or a scrambled siRNA. Experiments using these cells were performed 24 h after transfection. The siRNA sequences were as follows: siRNA CCN1-A, 5′-GGCAGACCCUGUGAAUAUATT-3′ (forward) and 5′- UAUAUUCACAGGGUCUGCCTT-3′ (reverse); siRNA CCN1-B, 5′-GUGGAGUUGACGAGAAACATT-3′ (forward) and 5′-UGUUUCUCGUCAACUCCACTT-3′ (reverse); siRNA integrin α_5_, 5′-CUCCUAUAUGUGACCAGAGTT-3′ (forward) and 5′-CUCUGGUCACAUAUAGGAGTT-3′ (reverse); siRNA integrin β_1_, 5′-GCCUUGCAUUACUGCUGAUTT-3′ (forward) and 5′-AUCAGCAGUAAUGCAAGGCTT-3′ (reverse); siRNA scrambled 5′-UUCUCCGAACGUGUCACGUTT-3′ (forward) and 5′-ACGUGACACGUUCGGAGAATT-3′ (reverse).

### 2.5. RNA Extraction and Reverse Transcriptase qPCR (RT-qPCR)

Total RNA was extracted from LO2 hepatocytes using TRIzol (Invitrogen, Carlsbad, CA). RT-qPCR was performed using the SYBR Green technique (Promega, Madison, WI, USA). GAPDH was used as an internal control. The siRNA sequences were as follows: murine CCN1, 5′-AACTGGCATCTCCACACGAG-3′ (forward) and 5′-ATCGGAACCGCATCTTCACA-3′ (reverse); human CCN1, 5′-CGCCTTGTGAAAGAAACCCG-3′ (forward) and 5′-GGTTCGGGGGATTTCTTGGT-3′ (reverse); murine integrin α_V_, 5′- GACTCCTGCTACCTCTGTGC-3′ (forward) and 5′-CTGGGTGGTGTTTGCTTTGG-3′ (reverse); murine integrin β_5_, 5′-AAGATTGGCTGGCGAAAGGA-3′ (forward) and 5′-TCTCTCCAAGCAAGGCAAGG-3′ (reverse); murine integrin α_M_, 5′-CCACACTAGCATCAAGGGCA-3′ (forward) and 5′-AAGAGCTTCACACTGCCACC-3′ (reverse); murine integrin β_2_, 5′-GAAGTGTCAGGACTTTACGACC-3′ (forward) and 5′-AGAGAGGACGCACCCGA-3′ (reverse); murine integrin α_5_, 5′-AGGGAAGAGCGGGCACTA-3′ (forward) and 5′-TTGGATGGAACTCAGGCAAC’ (reverse); murine integrin β_1_, 5′-ATGCCAAATCATGTGGAGAA -3′ (forward) and 5′-CATCGTGCAGAAGTAGGCATT-3′ (reverse); human integrin α_5_, 5′-CAGGGTCGGGGGCTTCAAC-3′ (forward) and 5′-CCCCAAGGACAGAGGTAGACA-3′ (reverse); human integrin β_1_, 5′-GAAGTCCGCACGCCG-3′ (forward) and 5′-TAAATTCATCTGCGCTTGCC-3′ (reverse); GAPDH, 5′- ACCACAGTCCATGCCATCAC-3′ (forward) and 5′-TCCACCACCCTGTTGCTGTA-3′ (reverse).

### 2.6. Immunoblotting and Immunoprecipitation 

Proteins were extracted in RIPA buffer supplemented with protease inhibitors. Then, protein concentrations were measured using the BCA protein assay. Immunoblotting was performed with appropriate primary antibodies and horseradish peroxidase conjugated secondary antibodies, followed by enhanced chemiluminescence detection.

For immunoprecipitation, cell lysates were incubated with the appropriate primary antibodies or control IgG at 4 °C overnight, followed by 1 h incubation with protein A/G-Sepharose beads. Immunoblotting was performed on immunoprecipitates using the appropriate antibodies.

### 2.7. ELISA

Concentrations of IL-1β and IL-18 in cultured medium of LO2 hepatocytes were measured using ELISA kits (Elabscience, Wuhan, China), according to the manufacturer’s instructions, and normalized to total protein.

### 2.8. Oil Red O Staining

LO2 hepatocytes were plated in a 12-well plate. After treatment, the LO2 cells were rinsed with PBS and fixed with 4% paraformaldehyde. The cells were stained with oil Red O for 30 min and rinsed with distilled water to remove excess staining solution. For quantification, Oil red O was dissolved in 60% isopropanol, and the absorbance was read at 540 nm.

### 2.9. Statistical Analysis

Results are reported as means ± SD and were analyzed using GraphPad Prism 8.0 software. The Student’s t-test was used to determine the significance of differences between two groups. Two-way analysis of variance, followed by Tukey’s post-hoc correction, was used for comparisons of more than two groups.

## 3. Results

### 3.1. CCN1 Is Upregulated in the Livers of Obese Mice and in FFA-Treated Hepatocytes

CCN1 expression was assessed in the livers of ND- or HFD-fed mice, and of db/m and db/db mice. The mRNA (Figure 1A) level of CCN1 was increased in HFD-fed mice compared with the ND group, and in db/db mice compared with db/m mice. CCN1 protein levels were elevated in both HFD (Figure 1B) and db/db mice (Figure 1C) compared with the respective control groups. To investigate whether high levels of FFA could regulate CCN1 expression in hepatocytes, LO2 cells were treated with FFA for 24 h. FFA increased the mRNA (Figure 1D) and protein (Figure 1E) levels of CCN1. These data demonstrate that CCN1 expression is upregulated in the livers of obese mice and in FFA-stimulated hepatocytes.

### 3.2. Activation of the NLRP3 Inflammasome by FFA Is Reduced by CCN1 Silencing

The upregulation of CCN1 expression due to high levels of FFA prompted us to evaluate the precise role of CCN1 in lipid-induced hepatic inflammation. LO2 hepatocytes were transfected with scrambled or CCN1-specific siRNAs. Both pairs of CCN1 siRNAs markedly reduced CCN1 protein levels compared with the scrambled siRNA, indicating their specificity (Figure 2A). Immunoblotting exhibited that CCN1 silencing suppressed the FFA-induced increase in cleaved caspase-1 levels, whereas the protein level of NLRP3 remained unchanged (Figure 2A). To further confirm the effect of CCN1 on the promotion of NLRP3 inflammasome activation by FFA, we tested the levels of IL-1β (Figure 2B) and IL-18 (Figure 2C) in the conditioned medium of LO2 using enzyme-linked immunosorbent assay (ELISA) and found that the concentrations of these two inflammatory cytokines increased following FFA treatment. CCN1 knockdown by siRNAs attenuated this increase. These results suggest that CCN1 could participate in the FFA-induced activation of the NLRP3 inflammasome in hepatocytes.

### 3.3. FFA-Induced Lipid Accumulation and Pyroptosis Are Attenuated by CCN1 Silencing in Hepatocytes

We next investigated the effects of CCN1 silencing on hepatocyte pathophysiology. LO2 cells were exposed to FFA. As shown in Figure 3A, both pairs of CCN1 siRNAs attenuated FFA-induced lipid accumulation compared with the scrambled siRNA. As long-duration FFA treatment is lipotoxic, and members of the GSDMD family cause membrane pore formation and lytic pro-inflammatory cell death [4], we first tested whether FFA-induced pyroptosis in hepatocytes was GSDMD-dependent. FFA treatment increased levels of GSDMD-NT, as well as those of HMGB1, an apoptotic factor released after cellular rupture [20]. The knockdown of CCN1 by two pairs of siRNAs attenuated the FFA-induced increases in GSDMD-NT and HMGB1 protein levels (Figure 3B). These results indicate that FFA promotes hepatocyte lipid accumulation and pyroptosis in a CCN1-dependent manner.

### 3.4. Integrin α_5_β_1_ Is Upregulated in the Liver of Obese Mice and FFA-Treated Hepatocytes

As CCN1 binds and signals through integrin receptors, we measured the levels of mRNAs of integrin α_V_β_5_, α_M_β_2_, and α_5_β_1_ and found that they were most abundant in the murine liver. The mRNA level of integrin α_V_β_5_ did not change in either HFD-fed or db/db mice compared with the respective control groups (Appendix A). Levels of integrin β_2_ remained unchanged, whereas those of integrin α_M_ mRNA were increased only in db/db mice (Appendix A).

The mRNA levels of integrin α_5_ were increased, whereas those of integrin β_1_ remained unchanged in the livers of both types of obese mice (Figure 4A). By contrast, both integrin α_5_ and β_1_ protein levels were increased in the livers of HFD-fed (Figure 4B) or db/db mice (Figure 4C) compared with the respective control groups. To confirm this result in vitro and investigate whether high levels of FFA could regulate the expression of integrin α_5_ or β_1_ in hepatocytes, LO2 cells were treated with FFA for 24 h. FFA increased the mRNA and protein levels of integrin α_5_ (Figure 4D,E). Similarly, the mRNA level of integrin β_1_ remained unchanged (Figure 4D), whereas its protein level was increased upon FFA stimulation (Figure 4E), indicating post-transcriptional regulation.

### 3.5. Binding of CCN1 to Integrin α_5_β_1_ Is Enhanced in Hepatocytes

The binding of CCN1 to integrin receptors is required for CCN1 signaling transduction [14]. Integrin α_5_β_1_ was the most upregulated integrin heterodimer in the FFA-treated hepatocytes and in the livers of obese mice, prompting us to study the ligation between CCN1 and integrin α_5_β_1_. Consistent with the upregulation of CCN1 and integrin α_5_ and β_1_ by FFA, the binding of CCN1 to integrin α_5_ or β_1_ was significantly enhanced in LO2 hepatocytes (Figure 5A). Similar results were obtained in obese (ND vs. HFD) mice in that the binding between CCN1 and integrin α_5_β_1_ was enhanced (Figure 5B). Although the mRNA level of integrin α_M_ was slightly increased in db/db mice (Appendix A), the binding between CCN1 and integrin α_M_β_2_ remained unchanged (Appendix A). These results suggest that integrin α_5_β_1_ might serve as a potential receptor of CCN1, leading to downstream signaling in the hepatocytes.

### 3.6. FFA-Activated NLRP3 Inflammasome Is Reduced by Integrin α_5_β_1_ Silencing 

The upregulation of integrin α_5_β_1_ under lipid-overload conditions prompted us to investigate its role in lipid-triggered inflammation. LO2 hepatocytes were transfected with scrambled siRNA, integrin α_5_ siRNA, integrin β_1_ siRNA, or a combination of integrin α_5_ and β_1_ siRNAs. All the non-control siRNAs showed markedly reduced levels of their target proteins compared with the levels in cells transfected with the scrambled siRNA, indicating their specificity (Figure 6A).

FFA did not affect NLRP3 protein levels with an increase in levels of cleaved caspase-1. The siRNA against integrin α_5_ or that against b1 attenuated the induction of cleaved caspase-1. The combination of siRNAs against integrins α_5_ and β_1_ abrogated FFA-triggered caspase-1 cleavage in LO2 hepatocytes (Figure 6A). To further confirm the effects of integrin α_5_β_1_ on FFA-promoted NLRP3 inflammasome activation, we examined the levels of IL-1β (Figure 6B) and IL-18 (Figure 6C) in the conditioned medium of LO2 using ELISA. Compared with the cells transfected with scrambled siRNA, the concentrations of these two inflammatory cytokines decreased following integrin α_5_, β_1_, or α_5_β_1_ knockdown.

### 3.7. FFA-Induced Lipid Accumulation and Pyroptosis Are Attenuated by Integrin α_5_β_1_ Silencing

We further investigated the effects of integrin α_5_β_1_ silencing on lipid accumulation and pro-inflammatory cell death. As shown in Figure 7A, both integrin α_5_ and β_1_ siRNA, or a combination of integrin α_5_β_1_ siRNA, attenuated FFA-induced hepatic lipid accumulation compared with the scrambled siRNA. As FFA-induced hepatocyte pyroptosis was CCN1-dependent, we investigated the participation of integrin α_5_β_1_, a potential receptor of CCN1, in FFA-induced pyroptosis. We found that both integrin α_5_ and β_1_ attenuated the induction of GSDMD-NT or HMGB1 by FFA. A combination of siRNA against integrin α_5_β_1_ was also effective in terms of GSDMD-NT and HMGB1 inhibition (Figure 7B). These results indicate that FFA promotes hepatocyte lipid accumulation, and that pyroptosis is integrin-α_5_β_1_-dependent.

## 4. Discussion

FFAs are saturated or unsaturated fatty acids containing 13–21 carbons. The levels of some fatty acids, such as OA and PA, are higher in the blood compared with others, and their levels may increase significantly in diabetes mellitus, obesity, and NAFLD [21,22,23]. Plasma concentrations range from 0.03 to 3.2 mmol/L for OA and 0.3 to 4.1 mmol/L for PA [24]. PA is the most abundant saturated fatty acid in plasma, which is known to stimulate pro-inflammatory responses in patients consuming an HFD. It is well-known that a diet with a high content of saturated fatty acids exerts a harmful effect by increasing the risk of multiple metabolic diseases, and the ratio of different dietary fatty acids could be more relevant than the individual amount of each fatty acid [25].

The NLRP3 inflammasome, which contains the NOD-like receptor NLRP3, the adaptor ASC, and the effector caspase-1, is the most studied and best characterized inflammasome regarding infection and in chronic inflammatory diseases. In addition to pathogen-associated molecular pattern ligands, the stimuli involved in NLRP3 activation include damage-associated molecular patterns, such as amyloid, uric acid crystals, and cholesterol crystals [26]. Excessive levels of FFA are lipotoxic and promote myocardial hypertrophy by triggering mitochondrial oxidative damage and mitochondrial DNA escape [27]. Endoplasmic reticulum stress-related cell damage is the main cause of renal and hepatic lipotoxicity [28,29]. Several studies have shown that lipid exposure results in NLRP3 inflammasome-mediated hepatic inflammation and glucose dysregulation in a high-fructose-induced NAFLD mouse model. Lipid exposure also aggravates hepatocellular lipid accumulation, as well as impairs insulin sensitivity by reducing JAK/STAT3 and PI3K signaling [30]. Assembly of the NLRP3 inflammasome and the production of pro-inflammatory cytokines are induced by mitochondrial reactive oxygen species in fatty hepatocytes [31,32]. In the current study, we aimed to elucidate the molecular mechanisms of lipid accumulation and pro-inflammatory cell death in FFA-treated hepatocytes.

CCN1 is strongly associated with NAFLD-related diseases such as diabetes and atherosclerosis. In addition, it is an exacerbating factor of fibrosis in NASH models [33]. HFD-fed mice and db/db mice become obese and develop hyperglycemia, making them susceptible to developing liver steatosis. Here, we found that CCN1 expression was upregulated in the livers of obese mice and in FFA-stimulated fatty hepatocytes. Bian et al. suggested that FFA induces CCN1 expression through the TLR4/MyD88/AP-1 pathway [9]. The pathogen-associated molecular and chronic inflammatory stimuli activate the NLRP3 inflammasome at different levels. The bacterial LPS transcriptionally upregulates NLRP3 via nuclear factor-kB, which is known as phase I activation. The second phase is the formation of a complex with ASC and the subsequent cleavage of pro-caspase-1 into its active form [4]. In contrast to the phase I activation of the NLRP3 inflammasome, in the present study, we found that the protein level of NLRP3 did not change upon FFA stimulation. A CCN1-specific siRNA only attenuated FFA-induced caspase-1 cleavage, without affecting NLRP3 protein levels, suggesting that CCN1 is a phase II regulator.

An increasing number of studies show that the activation of the NLRP3 inflammasome triggers the release of pro-inflammatory cytokines and initiates pyroptosis. When IL-1β and GSDMD are blocked, liver inflammation and extracellular matrix deposition are relieved in mice fed a high-sucrose diet or an HFD [34]. Generally, pyroptosis is accepted to be a caspase-1-dependent form of programmed cell death that is accompanied by the release of cytosolic contents, such as IL-1β, IL-18, and HMGB1, and even entire inflammasome complexes. HMGB1 release after inflammasome activation usually occurs after cellular rupture during various types of cell death, including pyroptosis, necroptosis, apoptosis, and ferroptosis [35]. Here, the CCN1 siRNA attenuated the FFA-induced increases in GSDMD-NT and HMGB1 protein levels, indicating that CCN1 participates in hepatic pyroptosis.

CCN1 is a highly regulated matricellular protein that regulates a wide range of cellular processes. It binds and signals through cell surface integrins, including α_6_β_1_, α_V_β_3_, and α_V_β_5_. Although CCN1 plays a critical part in hepatic inflammation, its integrin receptor has never been characterized. In our study, expression analysis using qRT-PCR and immunoblotting identified integrin α_5_ and β_1_ as the proteins most robustly and selectively induced in the livers of obese mice. A previous study reported that integrin α_5_β_1_ mediates adhesion to CCN1 and thrombin-stimulated DNA synthesis in astrocytoma cells [36]. Integrin β_1_ mediates the inflammatory response in several pathological processes, including M1 macrophage polarization in adipose tissue and angiopoietin-like protein 2-associated inflammation in chondrocytes [37,38]. Integrin β_1_ is a highly expressed integrin in hepatocytes and is actively enriched in lipotoxic extracellular vesicles. It enhances monocyte hepatic adhesion and infiltration, promoting diet-induced murine NASH [15]. A recent study demonstrated that integrin β_1_ activation in hepatocytes promotes intrahepatic lipid accumulation and NAFLD in mice by enhancing PKC α-stimulated fatty acid and triglyceride synthesis, as well as fatty acid uptake [39]. The results of the recent investigation by Campden et al. are consisting with our findings that integrin α_5_ mediates caspase-1 activation and the production of active IL-1β without affecting the transcription levels of NLRP3 inflammasome components [40]. Although the involvement of integrin α_5_β_1_ in the proliferation of hepatic stellate cells and hepatic carcinoma has been established [41,42], the role of integrin α_5_β_1_ in hepatocellular lipotoxicity and pro-inflammatory cell death, and the underlying molecular mechanisms, have not been previously been reported. Our co-immunoprecipitation experiments demonstrated an increase in binding between CCN1 and integrin α_5_β_1_ in hepatocytes following FFA stimulation and during obesity in mice, indicating integrin α_5_β_1_ as a potential receptor for CCN1. The knockdown of integrin α_5_β_1_ attenuated FFA-induced caspase-1 cleavage and excessive lipid accumulation, reducing HMGB1 and GSDMD-NT protein levels, indicating its participation in pro-inflammatory pyroptosis during obesity. Numerous factors and pathways have been reported to be involved in NAFLD/NASH models. Some of them have been correlated with the severity of the disease or proposed as possible biomarkers, including peroxisome proliferator-activated receptors, farnesoid X receptor, stearoyl coenzyme A desaturase 1, Fas, and G protein-coupled C-C chemokine receptors [43]. Our work demonstrated a partial participation of CCN1/integrin α_5_β_1_ in FFA-induced lipid accumulation and inflammatory pyroptosis. The crosstalk with these pathways needs to be further investigated. In addition, pharmacological treatments for chronic metabolic liver injury mainly focus on IL-1β and IL-1β receptors; there has been a lack of research on the potential of blocking other molecules in the inflammasome pathway. Integrin-based therapeutics have shown clinically significant benefits in patients with chronic inflammatory diseases [44]. In addition to its pro-inflammatory effects, CCN1/integrin α_5_β_1_ is an exacerbating factor in fibrosis and cell death in NASH model. Hence, more basic research and clinical trials involving the CCN1/integrin α_5_β_1_ pathway might provide insight in the future.

## 5. Conclusions

The present study provides evidence regarding the role of and mechanism by which the CCN1/integrin α_5_β_1_ may regulate hepatic lipid accumulation, inflammation, and pyroptosis. To our knowledge, this is the first identification of the CCN1 receptor in hepatocytes and the first report of the role of CCN1/integrin α_5_β_1_ in pyroptosis.

## Figures and Tables

**Figure 1 nutrients-14-03871-f001:**
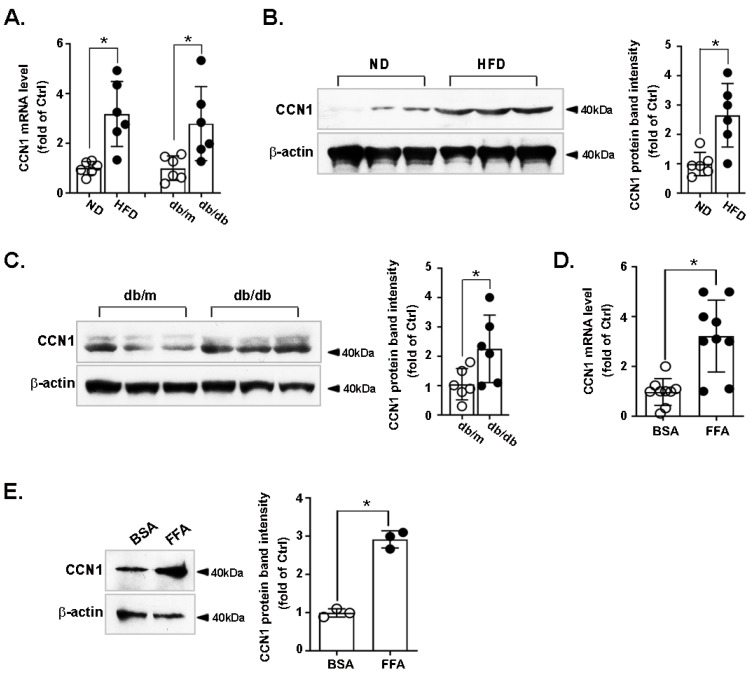
Expression of CCN1 in the livers of obese mice and in FFA-treated hepatocytes. Mice were fed with normal (ND) or high fat diet (HFD) for 16 weeks (n = 6 in each group) or were bred to be db/m or db/db mice (8 weeks old, n = 6 in each group). The mice were sacrificed at 16 weeks old or 8 weeks old, respectively. The livers were immediately dissected and subjected to quick-freezing in liquid nitrogen. The mRNA (**A**) and protein levels ((**B**): ND and HFD mice; (**C**): db/m and db/db mice) of CCN1 were measured by RT-qPCR and immunoblotting. LO2 hepatocytes were treated with bovine serum albumin (BSA) or free fatty acid (FFA, 1 mM) for 24 h. The CCN1 levels in cell lysates were analyzed using (**D**) RT-qPCR and (**E**) immunoblotting. The RT-qPCR data shown are from three independent experiments performed in triplicate. The immunoblots shown are representative of five independent experiments. Quantifications of band intensity were normalized to β-actin. * *p* < 0.05.

**Figure 2 nutrients-14-03871-f002:**
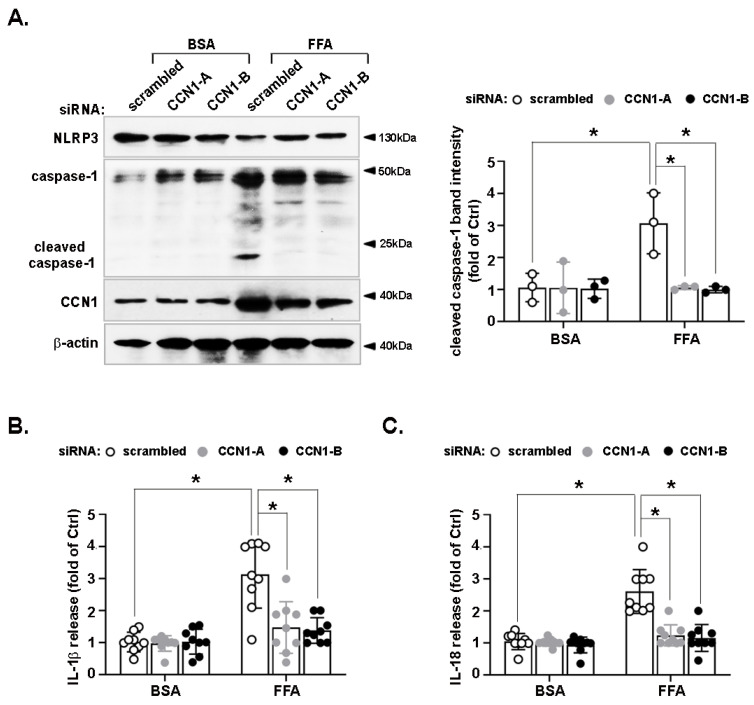
Effects of CCN1 on FFA-activated NLRP3 inflammasome in hepatocytes. LO2 hepatocytes were transfected with scrambled siRNA or two pairs of CCN1 siRNAs (CCN1-A and CCN1-B), then incubated with FFA (1 mM) for 24 h and ATP (5 mM for 30 min before harvest). Protein levels of NLRP3, caspase-1, CCN1, and β-actin in the cell lysates were analyzed by immunoblotting (**A**). Immunoblots shown are representative of three independent experiments. Quantifications of band intensity were normalized to β-actin. The concentrations of IL-1β (**B**) and IL-18 (**C**) in the conditioned medium were determined by ELISA. The ELISA data shown are from three independent experiments performed in triplicate. * *p* < 0.05.

**Figure 3 nutrients-14-03871-f003:**
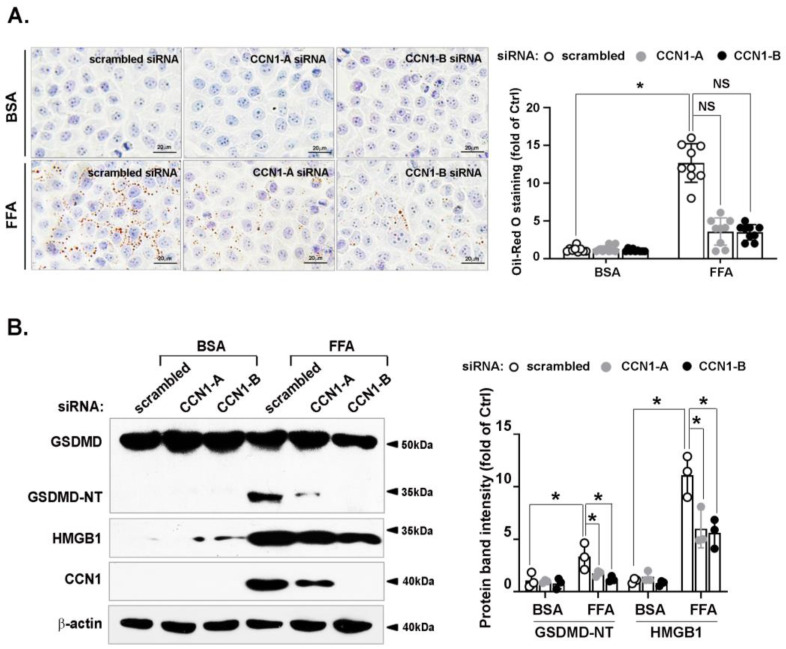
Effects of CCN1 on FFA-induced intracellular lipid accumulation and pyroptosis in hepatocytes. LO2 hepatocytes were transfected with a scrambled siRNA or two pairs of CCN1 siRNAs (CCN1-A and CCN1-B), then incubated with FFA (1 mM). (**A**) Prior to harvest, LO2 hepatocytes were stained with oil red O for 1 h. Images shown are representative of three independent experiments performed in triplicate (left). The quantification of lipid content was performed after isopropanol extraction (right). (**B**) Protein levels of GSDMD, HMGB1, CCN1, and β-actin in the cell lysates were analyzed by immunoblotting. Immunoblots shown are representative of three independent experiments. Quantifications of band intensity were normalized to β-actin. * *p* < 0.05, NS: not significant.

**Figure 4 nutrients-14-03871-f004:**
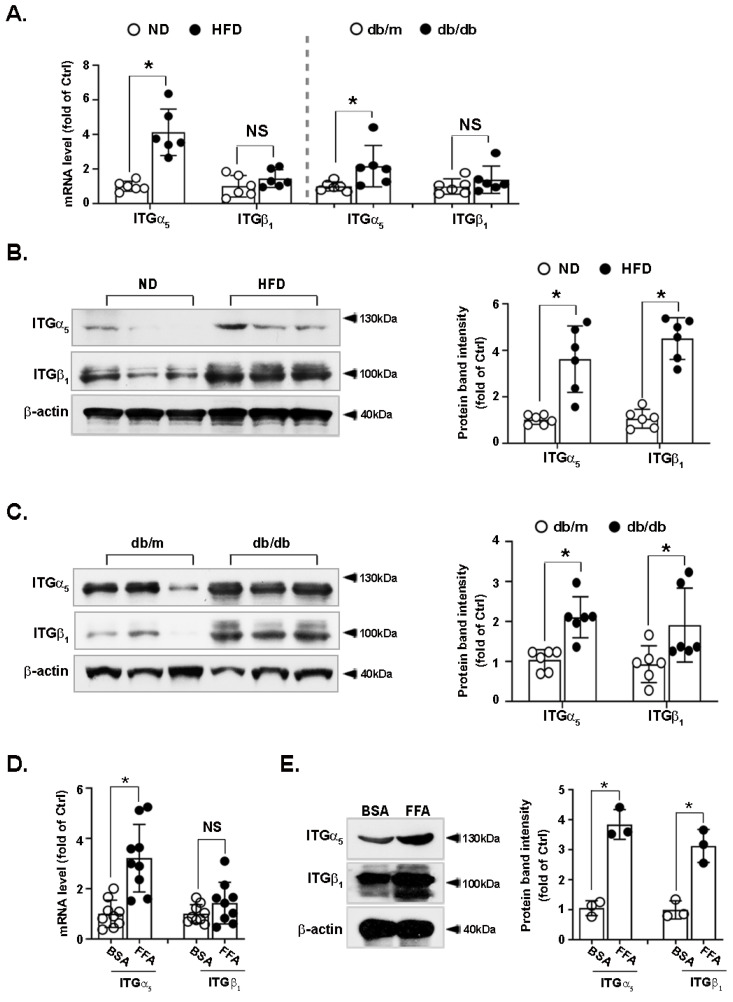
Expression of integrin α_5_ and β_1_ in the liver of obese mice and in FFA-treated hepatocytes. Mice fed a ND or HFD for 16 weeks (n = 6 in each group) and db/db and db/m mice (8 weeks old, n = 6 in each group) were sacrificed. The livers were immediately dissected and subjected to quick-freezing in liquid nitrogen. The mRNA levels of integrin α_5_ and β_1_ (**A**) were measured by RT-qPCR. The protein levels of integrin α_5_ and β_1_ in the livers of ND and HFD mice (**B**) and db/m and db/db mice (**C**) were measured by immunoblotting. LO2 hepatocytes were treated with FFA (1 mM) for 24 h. Levels of integrins α_5_ and β_1_ in the cell lysates were analyzed using (**D**) RT-qPCR and (**E**) immunoblotting. RT-qPCR data shown are from three independent experiments performed in triplicate. Immunoblots shown are representative of three independent experiments. Quantifications of band intensity were normalized to β-actin. * *p* < 0.05, NS: not significant. ITGα_5_: integrin α_5_; ITGβ_1_: integrin β_1_.

**Figure 5 nutrients-14-03871-f005:**
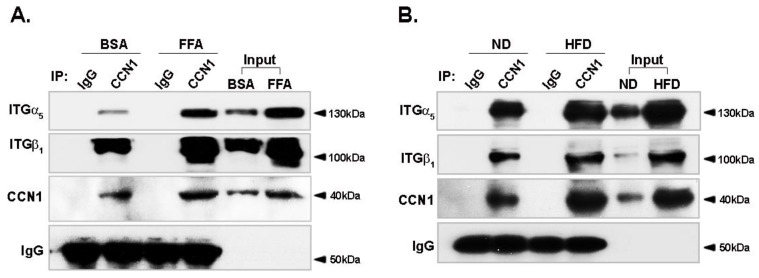
Effects of FFA on the binding of CCN1 to integrin a_5_β_1_. (**A**) Immunoblotting (IB; integrins α_5_ and β_1_) of whole cell lysates (input) and immunoprecipitates (IP: CCN1) from LO2 hepatocytes, with or without FFA (1 mM) treatment for 24 h. The immunoblots shown are representative of three independent experiments. (**B**) IB (integrins α_5_ and β_1_) of whole lysates (input) and IP (CCN1) from livers of db/m and db/db mice (8 weeks old). ITGα_5_: integrin α_5_; ITGβ_1_: integrin β_1_.

**Figure 6 nutrients-14-03871-f006:**
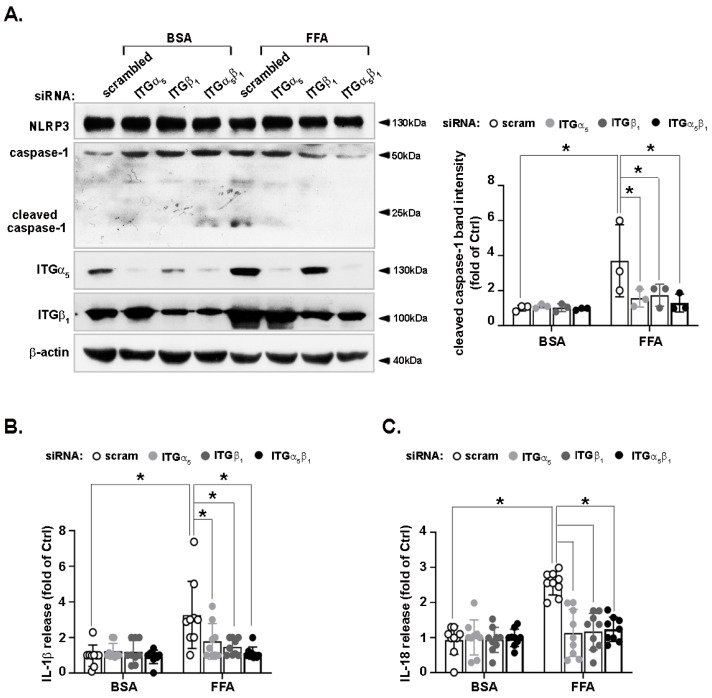
Effects of integrin α_5_β_1_ on the FFA-activated NLRP3 inflammasome in hepatocytes. LO2 hepatocytes were transfected with scrambled, integrin α_5_, integrin β_1_, or integrin α_5_ + β_1_ siRNA, then treated with FFA (1 mM) for 24 h and ATP (5 mM) for 30 min before harvest. Protein levels of NLRP3, caspase-1, CCN1, and β-actin in cell lysates were analyzed by immunoblotting (**A**). The immunoblots shown are representative of three independent experiments. Quantifications of band intensity were normalized to β-actin. The concentrations of IL-1β (**B**) and IL-18 (**C**) in the conditioned medium were determined by ELISA. ELISA data shown are from three independent experiments performed in triplicate. * *p* < 0.05. ITGα5: integrin α_5_; ITGβ_1_: integrinββ_1_.

**Figure 7 nutrients-14-03871-f007:**
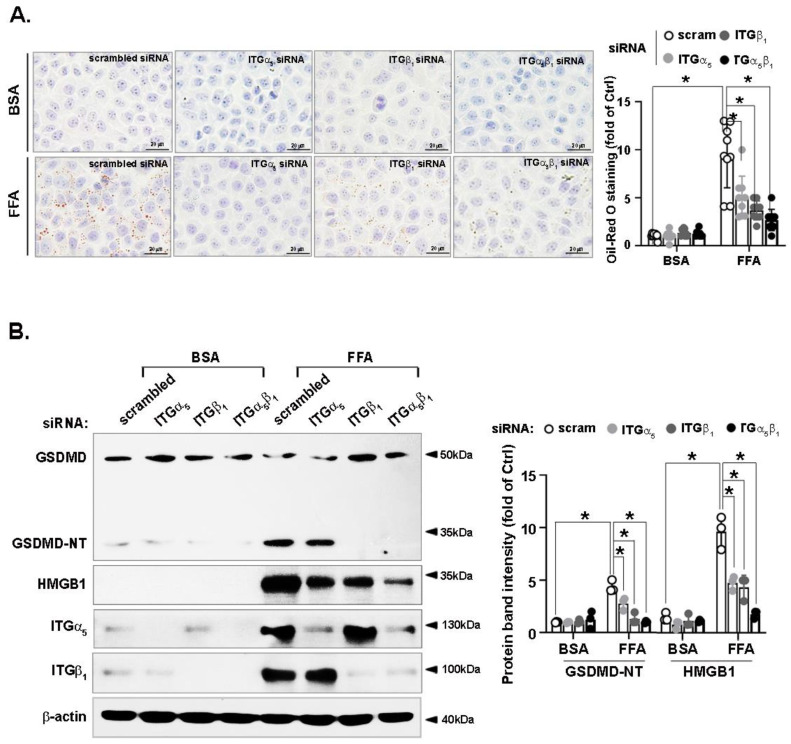
Effects of integrin α_5_β_1_ on FFA-induced lipid accumulation and pyroptosis in hepatocytes. LO2 hepatocytes were transfected with scrambled, integrin α_5_, integrin β_1_, or integrin α_5_ + β_1_ siRNA, then incubated with FFA (1 mM). (**A**) Prior to harvest, LO2 hepatocytes were stained with oil red O for 1 h. Images shown are representative of three independent experiments performed in triplicate (left). The quantification of the lipid was performed after isopropanol extraction (right). (**B**) Protein levels of GSDMD, HMGB1, CCN1, and β-actin in the cell lysates were analyzed by immunoblotting. The immunoblots shown are representative of three independent experiments. Quantifications of band intensity were normalized to β-actin. * *p* < 0.05. ITGα_5_: integrin α_5_; ITGβ_1_: integrin β_1_.

## Data Availability

All relevant data are provided within the manuscript and its Appendix A.

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
