# Peer review of "CCN1/Integrin α5β1 Instigates Free Fatty Acid-Induced Hepatocyte Lipid Accumulation and Pyroptosis through NLRP3 Inflammasome Activation"

_nutrients, 2022, doi:10.3390/nu14183871_

Round 1
Reviewer 1 Report
The Authors presented an interesting paper on inflammasome activation in NAFLDs scenario. The paper is quite well presented.
There are a few minor concerns.
In the Introduction, a few sentences (or a little paragraph) explaining the mechanism of pyroptosis would help the overall clarity of the study.
Materials and Methods: the "animal model" and the hepatocytes isolation sections should go first.
Some sentences sounds awkward, i.e.:"The upregulation of integrin was under lipid-overload conditions prompted us to investigate its role in lipid-triggered inflammation" (p. 8 line 289). Please check
A new paragraph can't start with "next"( i.e. p. 6 line 203, p.9 line 293).
The artwork are not good enough to be easy readable or clear.
Reviewer 2 Report
Yao et al. present an excellent study, where they unraveled a new mechanistic link between FFA and inflammation in hepatocytes. While the core of their experiments are conducted in-vitro, they elegantly apply their findings in-vivo. I only have a few questions on certain aspects of the manuscript and some advices to adapt the discussion section.
Major comments:
- CCN1 can be introduced in a bit more detail in the introduction; now, my feeling is that there is more information in the abstract than in the introduction.
- I am not sure whether it is due to my PC, but all figures are not of very high resolution… in order to proceed to publication, this will have to be improved
- Can the authors explain the different graph lay-out in figure 1 compared to figure 4? In figure 1 WT and db/db mice are shown in the same graph while in figure 4A both genotypes are depicted in separate graphs.
- A more general comment is related to the fitness of the manuscript to the journal Nutrients; while I think the manuscript is very well conducted, the link with Nutrients is not that apparent. Yes, FFA are the key trigger in the manuscript, but I would feel more comfortable if the authors could elaborate a bit more in the discussion on the composition of FFA (in order to make the manuscript more suitable for Nutrients)
- Small comment regarding the title of the manuscript: while the authors show that the integrin indeed instigates inflammation and lipid accumulation, they cannot claim that CCN1 is the only ligand that is responsible for this (other FFA-related ligands can also be involved). I would therefore advice to include in the discussion that while the current manuscript suggests a key involvement of CCN1, a potential contribution of other FFA-induced ligands cannot be ruled out.
o Nevertheless, a possible alternative approach that the authors might consider is considering figure 3 and adding both CCN1 siRNA and integrin siRNA together (with appropriate controls); such an experiment would indicate the contribution of CCN1 to integrin-related effects
- I am not familiar with the expression levels of integrinα5β1 in other tissues (though the authors indicate that they are mainly expressed in the liver). Is it possible that this integrin is also involved in mediating FFA-induced inflammatory responses in other organs? If this is possible, this is also something I would like to see in the discussion as such information adds relevance to the story line of the manuscript
Minor comments:
Line 156: small typo; twp-way should be two-way anova
Line 328: the claim that the in-vitro model can be considered a NAFLD cell model is a bit too far-fetched as the cell model only involves 1 cell type. I would rephrase this in such a way that it is more appropriate
